# Immune suppression in the early stage of COVID-19 disease

Wenmin Tian[1,2,11], Nan Zhang[1,2,11], Ronghua Jin[3,11], Yingmei Feng[3,11], Siyuan Wang[1,11], Shuaixin Gao[1], Ruqin Gao[4], Guizhen Wu[5], Di Tian[6], Wenjie Tan [5✉], Yang Chen [1,2✉], George Fu Gao [5,7✉] & Catherine C. L. Wong [1,2,8,9,10✉]

The outbreak of COVID-19 has become a worldwide pandemic. The pathogenesis of this infectious disease and how it differs from other drivers of pneumonia is unclear. Here we analyze urine samples from COVID-19 infection cases, healthy donors and non-COVID-19 pneumonia cases using quantitative proteomics. The molecular changes suggest that immunosuppression and tight junction impairment occur in the early stage of COVID-19 infection. Further subgrouping of COVID-19 patients into moderate and severe types shows that an activated immune response emerges in severely affected patients. We propose a two-stage mechanism of pathogenesis for this unusual viral infection. Our data advance our understanding of the clinical features of COVID-19 infections and provide a resource for future mechanistic and therapeutics studies.

[1] Center for Precision Medicine Multi-Omics Research, Peking University Health Science Center, Peking University, 100191 Beijing, China. [2] School of Basic Medical Sciences, Peking University Health Science Center, 100191 Beijing, China. [3] Beijing Youan Hospital, Capital Medical University, 100069 Beijing, China. [4] Qingdao Municipal Center for Disease Control and Prevention, Qingdao, China. [5] National Institute for Viral Disease Control and Prevention, Chinese Center for Disease Control and Prevention (China CDC), 102206 Beijing, China. [6] Center of Infectious Disease, Beijing Ditan Hospital, Capital Medical University, 100069 Beijing, China. [7] CAS Key Laboratory of Pathogenic Microbiology and Immunology, Institute of Microbiology, Chinese Academy of Sciences, 100101 Beijing, China. [8] Peking University First Hospital, 100034 Beijing, China. [9] Peking-Tsinghua Center for Life Sciences, 100871 Beijing, China. [10] Advanced Innovation Center for Human Brain Protection, Capital Medical University, 100069 Beijing, China. [11]These authors contributed equally: Wenmin Tian, Nan Zhang, Ronghua Jin, Yingmei Feng, Siyuan Wang. ✉email: tanwj@ivdc.chinacdc.cn; chenyang1816185048@bjmu.edu.cn; gaof@im.ac.cn; catherine_wong@bjmu.edu.cn

C OVID-19, the disease caused by infection with the virus SARS-CoV-2, has become a worldwide pandemic[1–5]. By 24 April 2020, there had been 2,649,859 confirmed cases, including 187,244 deaths, in 211 countries worldwide[6]. It is truly the first-time non-influenza pandemic. As yet, we know very little about this new virus and its pathogenesis[7]. At the beginning of the pandemic, research studies focused on the management and treatment of severe and critical patients[8]. Statistics showed that the elderly, especially those with underlying conditions such as heart disease, lung disease, obesity, and diabetes, have the most severe symptoms[9,10]. As the virus has now spread globally, more information is available about younger infected people in their 30s to 50s. In a growing number of cases, symptoms were expected to improve, but suddenly got worse. Patients can develop acute respiratory distress syndrome (ARDS) or even die suddenly in a short period of time. This sudden change implies a "two-stage" pattern of disease progression, but the underlying mechanisms are unknown.

Here, we applied a mass spectrometry-based, data-independent acquisition (DIA) quantitative proteomic approach to analyze urine samples from COVID-19 infection cases, healthy donors, and non-COVID-19 pneumonia cases (Fig. 1a). We found that immunosuppression and tight junction (TJ) impairment specifically occur in COVID-19 patients. Interestingly, we also found an activated immune response to some extent in the late stage of infection compared with the early stage. These data provide a map of molecular changes associated with the COVID-19 disease and provide hints as to how two-stage pathogenesis might occur.

## Results

**Urine proteome analysis of COVID-19 disease.** We collected 14 patients who tested positive for nasopharyngeal swab real-time PCR for SARS-CoV-2 infections, among which were six males and eight females with ages ranging from 30 to 77. We subdivided the 14 COVID-19 patients into nine with moderate disease type and five with severe disease type according to the information provided by the attending physicians at the hospitals where the samples were obtained (Supplementary Tables 1 and 2). The two control groups were 13 non-COVID-19 pneumonia patients and

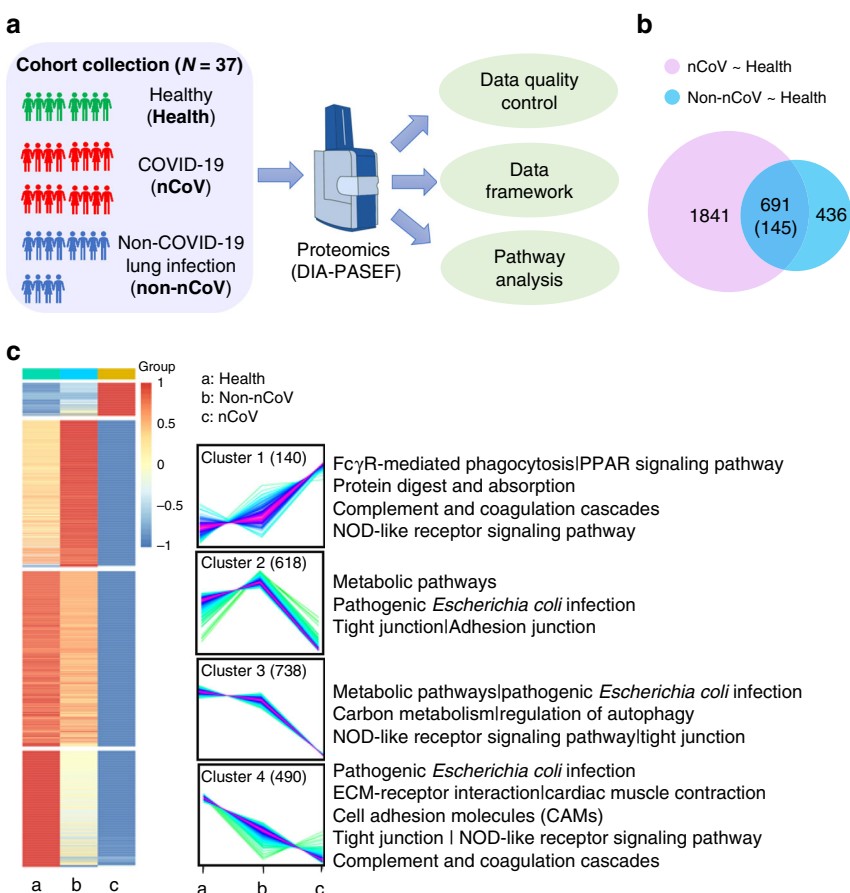

**Fig. 1 Quantitative urine proteomic studies. a** An illustration of the experimental design. A total of 37 urine samples were analyzed from three groups: healthy controls (10 samples); COVID-19 patients (14 samples); and non-COVID-19 pneumonia patients (13 samples). The data-independent acquisition (DIA) technique was applied for quantitative proteomics, which was performed by trapped ion mobility spectrometry coupled to TOF mass spectrometry (TIMS-TOF MS) with the parallel accumulation-serial fragmentation (PASEF) technique. Integrated data analysis involved protein expression, clustering, and functional correlational network strategies. **b** Venn diagram of significantly changed proteins (cut-off value of fold change >2 and fold change <0.5; p value (t test) <0.05) in the healthy control group compared to the COVID-19 patient group (pink) and to the non-COVID-19 pneumonia patient group (blue). A total of 1841 proteins were significantly changed specifically in COVID-19 patients compared to healthy controls. Six hundred and ninety-one proteins were changed in both disease groups. Among the 691 proteins, 145 proteins meet the condition of fold change >2 and fold change <0.5, p value (t test) <0.05. A total of 1986 proteins (the group of 1841 proteins and the group of 145 proteins) were used for functional analysis. **c** Proteomic changes in COVID-19 patients. Heatmap of 1986 proteins in COVID-19 patients in the comparison between healthy controls and non-COVID-19 patients. The color bar from red to blue represents the fold change from increasing to decreasing of all proteins identified in each group. Hierarchical clustering shows a clear group differentiation according to similarity. Numbers of proteins, intensity profiles, and selected enriched KEGG pathways are indicated for marked clusters. Color bar represents Z-score change from −1 to 1.

10 healthy people. The raw data were first processed using a double boundary Bayes imputation method and were standardized for further analysis (Supplementary Figs. 1 and 2 and Supplementary Data 1–3). Our data showed clear stratification among different cohorts according to the principal components analysis (PCA) (Supplementary Fig. 3 and Supplementary Data 3). For the data analysis, we firstly compared the three groups of COVID-19 infection cases, healthy donors, and non-COVID-19 pneumonia cases. A total of 5991 proteins were identified in all 37 samples. The levels of 1986 proteins were significantly changed in the COVID-19 group compared to the healthy donors and the non-COVID-19 pneumonia controls (Fig. 1b, Supplementary Fig. 4, and Supplementary Data 4–6). Surprisingly, we identified nearly ten times more down-regulated proteins than up-regulated ones in the COVID-19 group (Fig. 1c and Supplementary Data 7). KEGG (Kyoto Encyclopedia of Genes and Genomes) enrichment analysis revealed the molecular landscape associated with COVID-19 infections. More than ten pathways were significantly changed. In particular, COVID-19 had a strong impact on immune-related pathways, TJ pathways, and metabolic pathways (Fig. 1c and Supplementary Data 7).

**Immune system is suppressed in the early stage of COVID-19 disease**. We found that a large number of proteins associated with the immune response were down-regulated (Supplementary Fig. 5 and Supplementary Data 7). This suggests that the immune responses are suppressed in COVID-19 patients. We observed dramatically decreased protein levels of protein tyrosine phosphatase receptor type C, leptin, and tartrate-resistant acid phosphatase type 5, which are involved in lymphopenia, and platelet basic protein in COVID-19 patients (Fig. 2a, Supplementary Fig. 6, and Supplementary Data 7). These results are consistent with the lower lymphocyte and platelet counts in the blood tests of COVID-19 patients[11]. Complement C3, complement C1q subcomponent subunit C, complement C1r subcomponent, and PZP-like alpha-2-macroglobulin domain-containing protein 8 were down-regulated, which suggests the significant impairment of the complement system (Supplementary Fig. 7 and Supplementary Data 7). The level of spleen tyrosine-protein kinase, which is involved in FcγR-mediated phagocytosis, was dramatically decreased in patients, indicating that the phagocytosis of microphages, neutrophils, natural killer cells, and monocytes was suppressed (Fig. 2a, Supplementary Fig. 8, and Supplementary Data 7). A decreased level of apolipoprotein A-I in the serum has been reported during the transition of COVID-19 patients from mild to severe illness[12]. We also observed that COVID-19 patients showed the down-regulation of apolipoprotein A-IV (APOA4) and apolipoprotein E in COVID-19 patients, possibly associated with macrophage function.

Moreover, we observed decreased levels of proteins related to the chemokine signaling pathway (Supplementary Fig. 9 and Supplementary Data 7). C-C motif chemokine 14, C-C motif chemokine 18, and C-X-C motif chemokine ligand 12 were dramatically down-regulated in COVID-19 patients, indicating decreased monocyte activation, B cell migration, and T cell-mediated immune response (Fig. 2a and Supplementary Data 7). Neutrophil chemotactic factor C-X-C chemokine receptor type 2 and signal transducer and activator of transcription 3, 5B, and 6 were also down-regulated, which is suggestive of impaired cytokine production and degranulation of neutrophils, macrophages, and T lymphocytes (Supplementary Fig. 9 and Supplementary Data 7).

Acute hypoxia and ARDS are two of the major causes of the high case fatality rate in COVID-19 patients[13]. Previous research shows that a significant increase in the permeability of the alveolar epithelial barrier results in alveolar edema and exudate formation, and represents one of the major factors that contribute to the hypoxemia in ARDS[14]. The barrier function of the lung epithelium depends on a set of TJ heteromeric complexes, which seal the interface between adjacent epithelial cells[15]. TJs also exist between epithelial cells in other organs, such as intestine, kidney, and brain[16,17]. The disruption of TJ complexes is the major cause of epithelial barrier breakdown during virus infection[18]. We found that a number of proteins involved in TJ formation and cell–cell adhesion junctions were drastically down-regulated in COVID-19 patients, including TJ protein ZO-1, TJ protein ZO-2, claudin-2, claudin-3, claudin-11, claudin-19, Afadin, cingulin, protein crumbs homolog 3, cAMP-dependent protein kinase catalytic subunit alpha (PRKACA), and Rho GTPase-activating protein 17 were drastically down-regulated in COVID-19 patients (Fig. 2b, Supplementary Fig. 10, and Supplementary Data 7). This indicates that the virus may alter intercellular TJ formation and epithelial morphogenesis during viral invasion. This in turn may damage the physical barrier that protects the underlying tissues.

Among the significantly up- or down-regulated proteins in COVID-19 patient urine, metallothionein-1G, lipoprotein lipase, β2M, PRKACA, FOLR2, and APOA4, showed significant changes compared to both healthy controls and non-COVID-19 pneumonia cases (Fig. 2c). These proteins are potential biomarkers for differential diagnosis of COVID-19 to make it more precise. It is worth noting that four out of the six proteins are associated with the immune response and TJs, which are two featured pathways identified in COVID-19 patients.

**Immune response is activated in severe COVID-19 patients**. It is reported that a cytokine storm happens at the late stage of COVID-19 patients[19–21]. To understand the pathogenesis of COVID-19, it will be essential to find out how the transitions take place during the progression of the disease. To further investigate this, we subdivided the COVID-19 patients into nine moderate cases and five severe cases. PCA revealed a good separation of the moderate and severe COVID-19 patient samples (Supplementary Fig. 1c, d, Figs. 2c, d and 3c, d, and Supplementary Data 8 and 9). Gene Ontology enrichment analysis showed that most of the up-regulated proteins are involved in the complement and coagulation cascades, natural killer cell-mediated cytotoxicity, and platelet activation (Fig. 3a and Supplementary Data 10). Looking in detail the moderate and severe subgroups, we found, interestingly, that an activated immune response emerged to a certain extent in the late stage of the disease, while the immunosuppression effect remained in the early stage (Fig. 3b, c, Supplementary Figs. 11–13, and Supplementary Data 10 and 11). These results indicate that in the late stage of the disease the immune response was activated, which is consistent with an excessive immune response and cytokine storm in patients in severe and critical stages of COVID-19 patients[19–21]. Our study also identified two characteristic proteins, immunoglobulin lambda variable 3–25 and elongation factor 1-alpha 1, that may indicate the progression of the two stages of the COVID-19 disease. (Fig. 3d).

## Discussion

We applied the most advanced mass spectrometry technology to perform quantitative proteomics analysis of urine samples from COVID-19 patients and healthy controls and non-COVID-19 pneumonia patients. Several studies have shown that the compositions of proteins detected in urine samples can genuinely reflect the changes of the body health condition. The majority of urinary proteins originate from plasma components that pass through the glomerular filtration barrier, as well as liberated proteins from the kidney and urinary tract. Thus, in the absence of primary urological

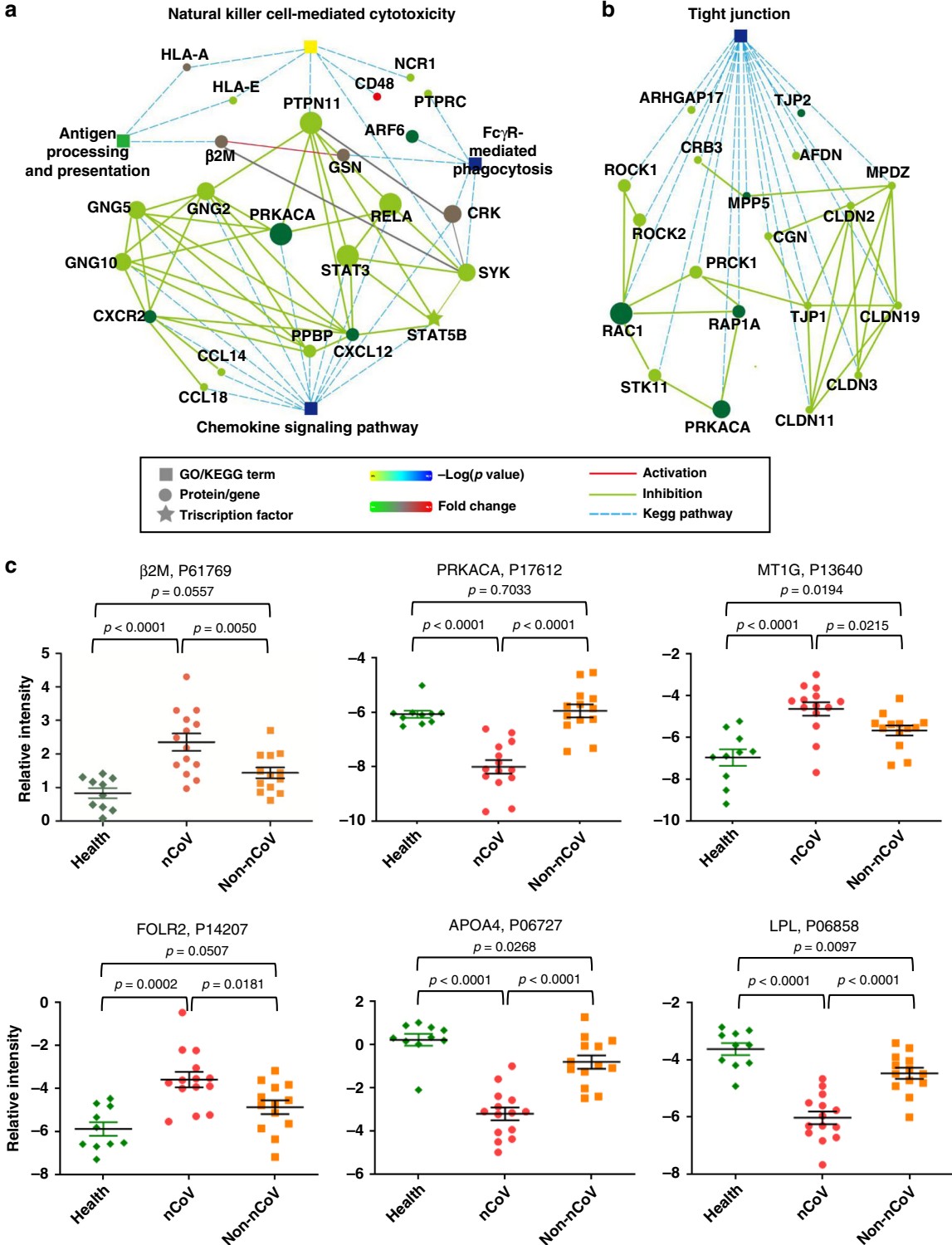

**Fig. 2 Functional analysis of 1986 specific proteins in COVID-19 patients. a** The interaction diagram of the FcγR-mediated phagocytosis, chemokine signaling pathway, natural killer cell-mediated cytotoxicity and antigen processing and presentation. Network nodes and edges represent proteins and protein–protein associations. Green solid lines represent inhibition; the red solid lines represent activation; the blue dotted lines represent the KEGG pathways. Color bar from red to green represents the fold change of protein level from increasing to decreasing. The significance of the pathways represented by −log($p$ value) (Fisher's exact test) was shown by color scales with dark blue as most significant. **b** The interaction diagram of proteins involved in tight junctions. Green solid lines represent inhibition; blue dotted lines represent the KEGG pathways. Color bar from red to green represents the fold change of protein level from increasing to decreasing. The significance of the pathways represented by −log($p$ value) (Fisher's exact test) was shown by color scales with dark blue as most significant. **c** The scatter plot graphs show six proteins, which are potential diagnostic markers for COVID-19. One-way ANOVA was used to analyze the data. For Health group, $n = 10$; for nCoV group, $n = 14$; for non-CoV group, $n = 13$. Data are presented as mean ± SEM. Source data are provided as a Source Data file.

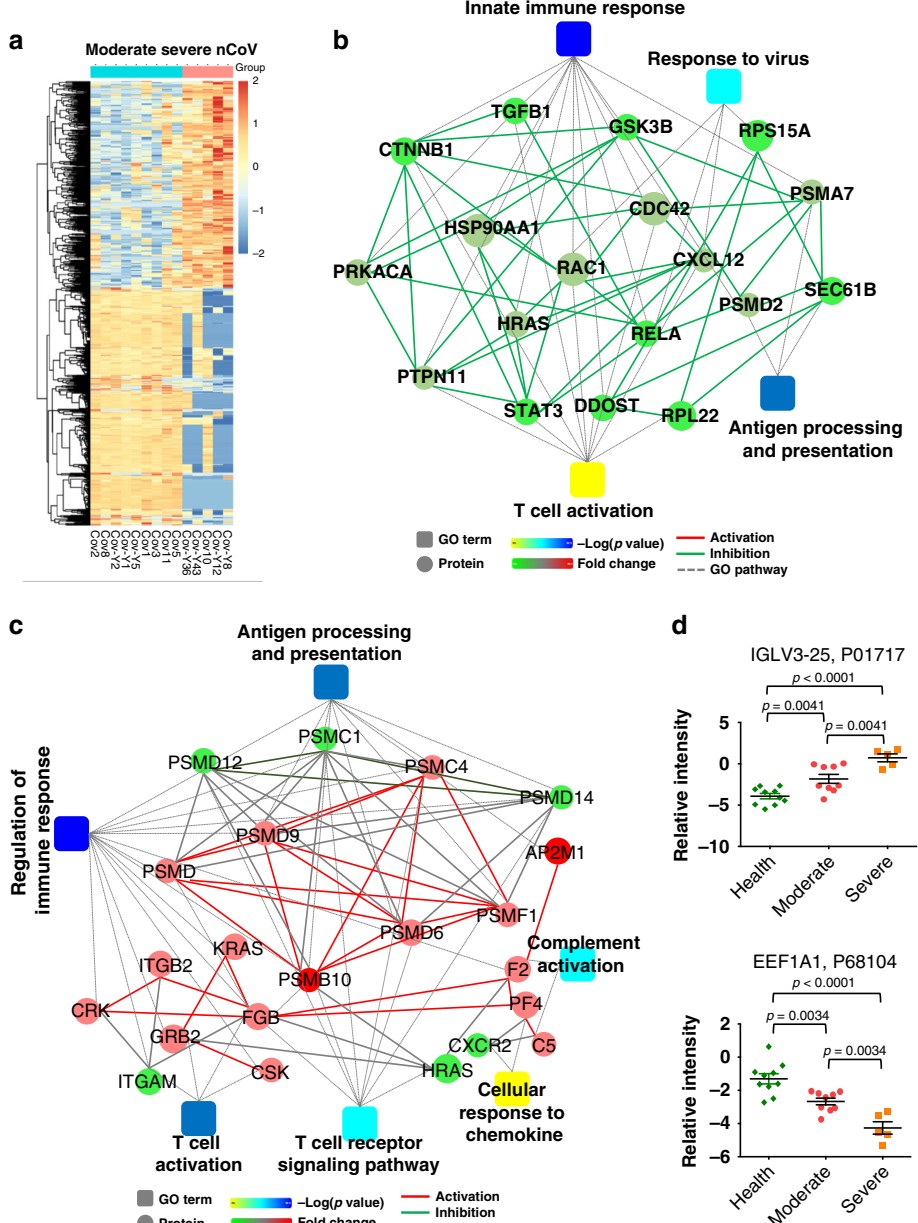

**Fig. 3 Immune system response in moderate and severe COVID-19 patients. a** Heatmap depicting the levels of differentially identified proteins in patients with moderate and severe COVID-19. The graphs show the relative intensity of differentially expressed proteins. Proteins included in the heatmap meet the requirement that fold change >2 or <0.5 and *p* value (*t* test) of <0.05 comparing severe to moderate patient samples. Color bar represents the relative intensity of identified proteins from −2 to 2. **b** The interaction diagram of proteins involved in the innate immune response, response to the virus, antigen processing and presentation, and T cell activation. Network nodes and edges represent proteins and protein–protein associations. Green solid lines represent inhibition; gray dotted lines represent GO pathways. Color bar from red to green represents the fold change of protein level from increasing to decreasing. The significance of the pathways represented by −log(*p* value) (Fisher's exact test) was shown by color scales with dark blue as most significant. **c** The interaction diagram of proteins involving in antigen processing and presentation, complement activation, cellular response to chemokine, regulation of immune response, T cell activation, and T cell receptor signaling pathway. Green solid lines represent inhibition; red solid lines represent activation; gray dotted lines represent GO pathways. Color bar from red to green represents the fold change of protein level from increasing to decreasing. The significance of the pathways represented by −log(*p* value) (Fisher's exact test) was shown by color scales with dark blue as most significant. **d** The scatter plot graphs showing two proteins that are potential diagnostic markers for severe COVID-19. One-way ANOVA was used to analyze the data. For Health group, *n* = 10; for nCoV Moderate group, *n* = 9; for nCoV Severe group, *n* = 5. Data are presented as mean ± SEM. Source data are provided as a Source Data file.

disease, the protein composition of urine is an appropriate mirror of general health status[22,23]. We uncovered changes in the protein landscape that revealed immunosuppression and impaired tight junctions in COVID-19 patients in the early stage of infection. Intriguingly, we also detected immune activation to a certain extent

in the late stage of infection (Fig. 4). These results will provide an important molecular basis for understanding the clinical symptoms of patients, and will shed light on how the two stage of the infection proceeds. Our data suggest that more attention should be paid to the dysregulation that occurs in the early onset of the infection. The

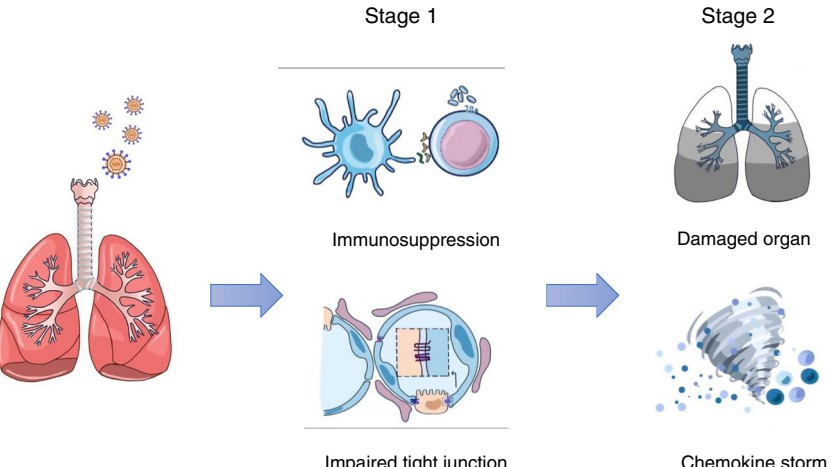

**Fig. 4 A "two-stage" model of COVID-19 pathogenesis.** The first stage of COVID-19 might involve suppression of the immune system and damage to tight junctions. The second stage might involve activated immune responses, contributing to cytokine storm and organ damage.

limitation of this study is that the dataset is correlative but not longitudinal. Nevertheless, these unusual features of COVID-19 during the course of human infections will guide us to further understanding of COVID-19 pathogenesis, mechanistic study, and clinical treatments.

## Methods

**Sample collection**. Urine samples were collected from Beijing Youan Hospital, Capital Medical University, and Chinese Center for Disease Control and Prevention between March 25 and April 10, 2020. Detailed information is shown in Supplementary Tables 1 and 2. COVID-19 patients were diagnosed according to the Chinese Government Diagnosis and Treatment guideline (Trial 5th Version) (Medicine 2020).

**Ethics statement**. Ethics approval was exempted by the institutional review board of the hospital as we collected and analyzed all data from the patients according to the policy issued by the National Health Commission of the People's Republic of China. Informed consent was obtained from all participants.

**Sample preparation for DIA analysis**. A measure of 200 µl urine was centrifuged at $6000 \times g$ for 10 min at 4 °C. The supernatant was precipitated by trichloroacetic acid solution at 4 °C for 4 h, and then centrifuged at $16,000 \times g$ for 30 min at 4 °C. After washing three times with acetone, the precipitate was dried with vacuum concentrator (Labconco, USA). The dried precipitate was resuspended in 40 µl 8 M urea in 500 mM Tris-HCl buffer (pH 8.5), incubated with 20 mM (2-carboxyethyl) phosphine hydrochloride (TCEP) (500 mM in 100 mM Tris/HCl pH 8.5) at room temperature for 20 min, and then alkylated with 40 mM iodoacetamide in the dark for 30 min. The mixture was diluted with 200 µl 100 mM Tris-HCl buffer (pH 8.5) to a final concentration of 1.3 M urea, followed by digestion with 3 µg trypsin protease to a final concentration of 0.0125 µg/µl at 37 °C for 16 h. Digestion was quenched by the addition of the formic acid (FA) at a final concentration of 5%. The sample was desalted using Monospin C18 column (GL Science, Tokyo, Japan).

The 1/3 volume of eluent was taken out and dried in vacuum. The peptides were re-dissolved with Milli-Q water and the concentration was measured using a BCA Peptide Assay Kit following the manufacturer's instructions.

The 2/3 remaining purified peptides were vacuum-centrifuged to dryness and reconstituted in Milli-Q water with 0.1 vol% FA for liquid chromatography-mass spectrometry analysis. For DIA experiments, iRT (indexed retention time) calibration peptides were spiked into the sample.

**Sample preparation for spectral library**. Eighty-eight micrograms of mixed protein from 10 healthy and 11 COVID-19 patients were processed as above and the added enzyme to protein ratio is 1:50. Purified peptides were reconstituted in 80 µl fraction buffer A (98% $H_2O$, 2% acetonitrile).

**High-pH reversed-phase fraction**. Approximately 88 µg mixed peptides were fractioned on a Chromatographic column (BEH C18, 300 Å, 1.7 µm, 1 mm × 150 mm) coupled to a Waters Xevo$^{TM}$ ACQUITY UPLC (Waters, USA) within 80 min gradient and concatenated into 62 fractions. The first fraction is mixed with the last fraction, and the rest is mixed with two fractions every 30 fractions

sequentially. Finally, 31 fractions were obtained. All fractions were vacuum-centrifuged to dryness and reconstituted in 10 µl Milli-Q water with 0.1 vol% FA. iRT peptides were spiked before data-dependent acquisition (DDA) analysis.

**Liquid chromatography**. We employed a nanoElute liquid chromatography system (Bruker Daltonics). Peptides (200 ng of digest) were separated within 90 min at a flow rate of 300 nL/min on a 25 cm × 75 µm column with a laser-pulled electrospray emitter packed with 1.5 µm ReproSil-Pur 120 C18-AQ particles (Dr. Maisch). Mobile phases A and B were water and acetonitrile with 0.1 vol% FA, respectively. The %B was linearly increased from 2 to 22% within 70 min, followed by an increase to 37% within 8 min and a further increase to 95% within 5 min before the last 7 min 95% process.

**Mass spectrometry**. All 31 fraction samples were analyzed on a hybrid trapped ion mobility spectrometry (TIMS) quadrupole time-of-flight mass spectrometer (MS) (TIMS-TOF Pro, Bruker Daltonics) via a CaptiveSpray nano-electrospray ion source. The MS was operated in data-dependent mode for the ion mobility-enhanced spectral library generation. We set the accumulation and ramp time was 100 ms each and recorded mass spectra in the range from $m/z$ 100–1700 in positive electrospray mode. The ion mobility was scanned from 0.6 to 1.6 Vs/cm$^2$. The overall acquisition cycle of 1.16 s comprised one full TIMS-MS scan and 10 parallel accumulation-serial fragmentation (PASEF) MS/MS scans. When performing DIA, we define quadrupole isolation windows as a function of the TIMS scan time to achieve seamless and synchronous ramps for all applied voltages. We defined up to eight windows for single 100 ms TIMS scans according to the $m/z$-ion mobility plane. During PASEF MSMS scanning, the collision energy was ramped linearly as a function of the mobility from 59 eV at $1/K0 = 1.6$ Vs/cm$^2$ to 20 eV at $1/K0 = 0.6$ Vs/cm$^2$.

**Generation of spectral library and DIA-PASEF processing**. Raw files were processed using a developmental version of Spectronaut (v14.0.200409.43655, Biognosys). The ion mobility-enhanced library was generated from DDA-PASEF raw data using Spectronaut's Pulsar database search engine with 1% false discovery rate control at peptide-spectrum match, peptide, and protein level. Carbamidomethyl (C) was set as fixed modifications, and oxidation (M) and acetyl (protein N-term) were set as variable modifications. For the subsequent targeted analysis of DIA-PASEF data, DIA files were processed using Spectronaut with default settings, but the correction factor of XIC IM extraction window was set to 0.8 instead of 1.0. $Q$ values at the precursor and protein level were set to <1%. Each patient sample is treated as a biological duplicate. A quality control sample of mixed aliquots from each sample was applied for every four sample run. The median coefficient of variation for quantification was 18.6% on the protein level after median normalization.

**Statistical analysis**. To impute the proteomic data, we first used locally weighted polynomial regression[24] (lowess in R v.3.6.3[25]) to compute the local polynomial fit for protein number and protein-detecting rate in each stage(time point). Two boundary thresholds, 0.15 and 0.5, were used to separate the data into three parts. When a protein-detecting rate is <0.15, it is probably because the detected value is due to a technical error. For these proteins, no imputation was applied. When a protein-detecting rate is >0.5, the missing value was probably due to the detection accuracy limitation of the LC/MS. In this case, the missing value was replaced with a median value. When a protein-detecting rate is between 0.15 and 0.5, it is probably because the protein expression is unstable for detection.

In this case, we first calculated the missing probability of a protein using Bayes theory,

$$\text{missp} = PA(PBA/((PBA \times PA) + (0.05 \times (1 - PA))))$$

where PBA is the group missing rate and PA the total missing rate of each protein.

Then, we determined the predicted imputation number (IN) of each protein in each group,

$$IN = (1 - \text{missp})Mi$$

where Mi is the number of undetected sample number of a protein in group i.

Finally, the random method was used to determine the samples to be imputed. The imputation value[15] was then defined by,

$$IV = \min(Mi/2, IN)$$

Imputed data were then normalized using LogNorm algorithm. PCA (muma v1.4 package, https://www.rdocumentation.org/packages/muma) and fastcluster v.1.1.1 (https://www.rdocumentation.org/packages/fastcluster/versions/1.1.25/) using euclidean distance was used to perform the clustering analysis of samples.

R package Genefilter (https://www.rdocumentation.org/packages/genefilter/versions/1.54.2) was used in the calculation of the fold-change values of proteins. Fold change of 2, fold change <0.5, and p value (t test) of 0.05 were used to filter differential expression proteins.

Mfuzz v.2.46.0 (https://www.bioconductor.org/packages/release/bioc/html/Mfuzz.html) was used to detect different sub-clustering models of gene expression among groups. R v.3.6.3[26] was used to implement Fisher's exact test. String version 11[27] was used for protein–protein interaction network analysis.

The KEGG ligand database was used to obtain the compound and enzyme relationship. Venn diagram, heatmap, and network visualization were performed using the ggplot2[28] packages and Cytoscape v.3.5.1[29] implemented in the omicsbean workbench. Ingenuity pathway analysis was performed to explore the downstream effect in significant regulated proteins dataset. The z-score algorithm was used to predict the activation state (either activated or inhibited)[30] of the biological process. If the z-score $\leq -2$, the process is predicted to be statistically significantly inhibited.

**Reporting summary**. Further information on research design is available in the Nature Research Reporting Summary linked to this article.

## Data availability

Reference FASTA files contain human UNIPROT database (only reviewed entries) (human 20,421 entries, downloaded July 2019) and SARS-CoV-2 Uniprot database, which combined with SARS protein database (38 viral entries, April 2020). Latest GO database[31] (https://www.ebi.ac.uk/QuickGO/) and KEGG pathway database[32] (https://www.kegg.jp/kegg/pathway.html) were used for gene ontology and pathway enrichment analysis. The KEGG ligand database (https://www.kegg.jp/kegg/ligand.html) was used to obtain the compound and enzyme relationship. Data that support the findings of this study have been deposited in iProX (integrated proteome resources) ProteomeXchange under the accession code PXD020522. Source data are provided with this paper.

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

## Acknowledgements

We thank Bruker Daltonics Inc. for their support in proteomic analysis. We are grateful to OmicsBean (Gene For Health Inc.) for their assistance in data analysis. We also thank SpectronautTM (Biognosys Inc.) for their support in database searching. We would like to thank Wenting Li for her contribution in data processing and figure plotting; Feng Shao (National Institute of Biological Sciences, Beijing), Zhihong Liu (Zhengjiang University), Chengyu Jiang (Peking Union Medical College), Chenqi Xu (Shanghai Institute of Biochemistry and Cell Biology, Chinese Academy of Sciences), Minghui Zhao (Peking University First Hospital), Wenke Han (Peking University First Hospital), Youhe Gao (Institute of Basic Medical Sciences, Chinese Academy of Medical Sciences), and Zhihua Chen (The Second Affiliated Hospital, Zhejiang University School of Medicine) for their valuable comments to this study. The authors are grateful to Xue Sun, Hui Sun, Cuitong He, Rui Ji, and other members from Center for Precision Medicine Multi-omics Research for their assistance on information collection. This work is supported by the Fundamental Research Funds for the Central Universities (BMU2017YJ003, BMU2018XTZ002) and Research Funds from Health@InnoHK Program launched by Innovation Technology Commission of the Hong Kong Special Administrative Region, the PKU-Baidu Fund 2019BD007, Training Program of the Big Science Strategy Plan 2020YFE0202200 and National Program on Key Research Project of China (2016YFD0500301).

## Author contributions

C.C.L.W. and G.F.G. conceived the project. C.C.L.W. supervised mass spectrometry proteomics experiments. W.T., R.J., Y.F., G.W., R.G., and D.T. collected clinical samples.

N.Z. and S.G. performed proteomics experiments. C.C.L.W., Y.C., W.T., and S.W. analyzed the data. C.C.L.W., G.F.G., Y.C., and W.T. wrote the manuscript.

## Competing interests

The authors declare no competing interests.
