## [Peer Review File · Nature Communications]

REVIEWER COMMENTS

Reviewer #1 (Remarks to the Author):

This is a small and intriguing study of urine proteomics from patients with mild-moderate COVID-19. Compared to healthy controls and those without pneumonia, many proteins were downregulated in the urine of those with COVID-19. Many of the results are quite provocative, with notable downregulation of proteins associated with the immune system and tight junctions. However, I have a few big picture comments, as noted below:

1. Additional clinical characteristics about the COVID patients and controls would be essential to understanding the meaning and impact of these findings. For example, what was the level of kidney function? Was there measurable proteinuria? How many, if any, of these mild-moderate patients progressed to severe-critical disease?
2. Does urinary protein excretion of these molecules suggest that there are lower levels of expression systemically or in renal epithelial cells?

Reviewer #2 (Remarks to the Author):

Review:

The manuscript entitled "Immune-suppression and tight junction damage of COVID-19 infection at early stage" describes the differences identified between proteins that are present in the urine of covid19 positive patients in comparison to patients with pneumonia (non covid19) and healthy controls. One third of proteins is altered in abundance in covid19 urine samples in comparison to controls and the covid19 group included 10 times more down regulated proteins than controls. A molecular pathway analysis highlights tight junctions and immune response. The authors select eight potential diagnosis markers for covid19.

Overall the manuscript is concise and provides detailed information for every step during sample acquisition, data processing, and data analysis. For publication the authors should consider to include the following:

- (1) A link to a data repository that provides the raw data used in this study. This is especially important given that the data shows a strong bias towards proteomic deviations in the covid19 sample cohort.
- (2) Three different sample patient cohorts were used in the study. A differential analysis of the pneumonia patient group in the absence of covid-19 infection is missing. For example, what disease markers previously described for pneumonia in urine were recapitulated in this group and are present or absent in covid19 patients?
- (3) Urine represents a highly selective filtrate of blood as a consequence of the activity of nephron activity. What markers were included to ensure that an altered proteomic profile in covid19 patients is or is not a consequence of an altered functioning of the nephrons? If available, the authors should report and correlate the proteomic data obtained to data that are a result of a standard clinical analysis of the urine. Specifically, are marker proteins that indicate normal functioning of nephrons affected during covid19 infection?
- (4) The authors are carefully proposing a 2-stage model for covid19 progression based on the results obtained. The model lacks consistency in the sense that the first stage reflects molecular alterations (immune suppression and impaired tight junction) whereas the second stage reflects organ and organism wide changes. Thus it is unclear whether it is really two stages of disease

progression or merely a linear amplification of the molecular alterations that finally influence the functioning of whole organs. Indeed, specific alterations between early and late stage covid19 urine samples would be needed to support any two stage model of disease.

Reviewer #3 (Remarks to the Author):

This was a focused proteomics analysis of urine from three classes of subjects: healthy subjects, COVID-19 patients and patients with non-COVID pneumonia. Based on proteomics combined with KEGG analysis, they found that in COVID patients there was protein content signature that suggested early suppression of the immune response (Figure 2A) and that there was an enhancement of tight junction proteins present in urine (Figure 2B). Based on their analysis, they determined that COVID-19 patients showed hallmark changes of a set of 8 proteins associated with a COVID-specific response (e.g. increased β 2M, TRIM4; decreased PRKACA, LPL), which may have diagnostic value.

The authors proposed that the protein expression profile suggests that "the COVID-19 virus may alter intercellular TJ formation and epithelial morphogenesis during viral invasion through the human airway." (Lines 149-151). However, with the exception of claudin-3, the claudin profile they identified from the urine samples (claudin-2, claudin-11, claudin-19) does not match the expected claudin expression profile if these proteins were to have originated in the lung epithelium (e.g. high claudin-4, claudin-7, claudin-18). Based on this it is not clear whether the proteins represent damaged lung epithelial cells or whether they are due to damage from other cells.

Another concern is that the samples were combined COVID-19 patients with mild-to-moderate stage disease; it would strengthen the study to stratify these samples into two different groups and to also analyze severe stage COVID-19 patients. Also, the type of pneumonia of the non-COVID patients was not clear (e.g. bacterial vs viral) and was also combined. Although the PCA in Figure S3 analysis suggests that the three different groups of patients do not overlap, there may still be subgroups within each set of subjects.

Although the proteomics analysis was interesting, the study as it currently stands is limited and does not provide experimental validation of the model of the progression of COVID-19 proposed in Figure 3.

Reviewers' Comments:

We thank all three reviewers for their careful analyses of the manuscript and instructive comments. We have carefully revised the manuscript to fully address all concerns.

Reviewer #1

This is a small and intriguing study of urine proteomics from patients with mild-moderate COVID-19. Compared to healthy controls and those without pneumonia, many proteins were downregulated in the urine of those with COVID-19. Many of the results are quite provocative, with notable downregulation of proteins associated with the immune system and tight junctions. However, I have a few big picture comments, as noted below:

We sincerely appreciate the reviewer's recognition of our work.

1. Additional clinical characteristics about the COVID patients and controls would be essential to understanding the meaning and impact of these findings. For example, what was the level of kidney function? Was there measurable proteinuria? How many, if any, of these mild-moderate patients progressed to severe-critical disease?

We thank the reviewer for this comment, which we agree with. The clinical characteristics of COVID-19 patients have been listed in the updated Table S1. We also revised the text in paragraph two, line 6-72, to *"We subdivided the 14 COVID-19 patients into 9 with moderate disease type and 5 with severe disease type according to the information provided by the attending physicians at the hospitals where the samples were obtained (Table S1)."*

Urine test results showed no proteinuria in six patients and mild (+) proteinuria in two patients. The clinical results showed no indication of renal impairment. No urine test results were available for the other six patients. These six samples came from a hospital in Wuhan. Under the emergency prevention and control of the pandemic, COVID-19 patients were admitted to the Department of Respiratory and Infectious Diseases and only viral nucleic acid, blood test and CT were required. Though we lack of some clinical data, we examined the expression level of albumin, hemoglobin and myoglobin in our proteome results. The presence of these proteins in the urine often indicates proteinuria of nephrotic patients. We found that none of these three proteins were significantly changed in urine samples.

Large amount of urinary proteins is one of the clinical symptoms of kidney function damage. Therefore, when proteomics is used to study kidney disease, a depletion of high abundant proteins is necessary during urine sample preparation process in order to avoid ion suppression effect. Even so, a large variation often appears in the results of protein identification. In our study, all proteomics experiments were done without

depletion process and we identified about 2200 protein ID with a constant loading amounts of 300 ng for each sample. These results indicated that there is no ion suppression effect thus depletion for high abundant proteins are not necessary for our samples.

Taking these findings together, we conclude that there is no significant renal function loss in COVID-19 patients.

Unfortunately, we lack any information about the progression of the patients.

2. Does urinary protein excretion of these molecules suggest that there are lower levels of expression systemically or in renal epithelial cells?

We thank the reviewer for these insightful questions.

Several studies have shown that the compositions of proteins detected in urine samples can genuinely reflect the changes in health conditions[1, 2]. The majority of urinary proteins originate from plasma components that pass through the glomerular filtration barrier, as well as proteins liberated from the kidney and urinary tract. Thus, in the absence of primary urological disease, marked changes of cellular proteins are an important early warning signal for disease and the protein composition of urine is an appropriate mirror of general health status. Urinary proteomic studies have been widely used to identify many candidate biomarkers for various diseases, such as acute kidney injury[3, 4], bladder cancer[5], diabetic nephropathy[6], muscular dystrophy[7], Kawasaki disease[8, 9] and autoimmune myocarditis[10]. The advantage of using urine as a source of clinical marker molecules is the fact that urine samples can be obtained non-invasively and be sampled in a continuous way. In this respect, the increased usage of urine proteomics has greatly improved the scope of the bioanalytical analysis of diseases.

References:

1. Zhao, M., et al., *A comprehensive analysis and annotation of human normal urinary proteome*. *Sci Rep*, 2017. **7**(1): p. 3024.
2. Wu, J. and Y. Gao, *Physiological conditions can be reflected in human urine proteome and metabolome*. *Expert Rev Proteomics*, 2015. **12**(6): p. 623-36.
3. Zurbig, P., J. Siwy, and H. Mischak, *Emerging urine-based proteomic biomarkers as valuable tools in the management of chronic kidney disease*. *Expert Rev Mol Diagn*, 2019. **19**(10): p. 853-856.
4. Ho, J., et al., *Mass spectrometry-based proteomic analysis of urine in acute kidney injury following cardiopulmonary bypass: a nested case-control study*. *Am J Kidney Dis*, 2009. **53**(4): p. 584-95.
5. Majewski, T., et al., *Detection of bladder cancer using proteomic profiling of urine sediments*. *PLoS One*, 2012. **7**(8): p. e42452.

6. Zubiri, I., et al., *Diabetic nephropathy induces changes in the proteome of human urinary exosomes as revealed by label-free comparative analysis*. J Proteomics, 2014. **96**: p. 92-102.
7. Gargan, S., et al., *Identification of marker proteins of muscular dystrophy in the urine proteome from the mdx-4cv model of dystrophinopathy*. Mol Omics, 2020. **16**(3): p. 268-278.
8. Kentsis, A., et al., *Urine proteomics for discovery of improved diagnostic markers of Kawasaki disease*. EMBO Mol Med, 2013. **5**(2): p. 210-20.
9. Hu, H.M., et al., *New biomarkers of Kawasaki disease identified by urine proteomic analysis*. FEBS Open Bio, 2019. **9**(2): p. 265-275.
10. Zhao, M., et al., *Urinary candidate biomarkers in an experimental autoimmune myocarditis rat model*. J Proteomics, 2018. **179**: p. 71-79.

Reviewer #2

The manuscript entitled "Immune-suppression and tight junction damage of COVID-19 infection at early stage" describes the differences identified between proteins that are present in the urine of covid19 positive patients in comparison to patients with pneumonia (non covid19) and healthy controls. One third of proteins is altered in abundance in covid19 urine samples in comparison to controls and the covid19 group included 10 times more down regulated proteins than controls. A molecular pathway analysis highlights tight junctions and immune response. The authors select eight potential diagnosis markers for covid19.

Overall the manuscript is concise and provides detailed information for every step during sample acquisition, data processing, and data analysis. For publication the authors should consider to include the following:

We sincerely appreciate the reviewer's recognition of our work.

(1) A link to a data repository that provides the raw data used in this study. This is especially important given that the data shows a strong bias towards proteomic deviations in the covid19 sample cohort.

We thank the reviewer for the important suggestion. The data which include raw files and the intact results of database searches will be uploaded onto the website below, after the review process is finished.

<https://www.iprox.org/page/SSV024.html?url=1595493850892iFyN>

(2) Three different sample patient cohorts were used in the study. A differential analysis of the pneumonia patient group in the absence of covid-19 infection is missing. For example, what disease markers previously described for pneumonia in

urine were recapitulated in this group and are present or absent in covid19 patients?

We thank the reviewer for this insightful question. We have systematically searched the literature and found that specific biomarkers have been published for diagnosis of pneumonia, especially community-acquired pneumonia (CAP). Procalcitonin (PCT) and C-reactive protein (CRP) are the most widely used biomarkers. New biomarkers like monocyte HLA-DR and soluble CD14 (presepsin), interleukin-6 (IL-6), prohormones such as adrenomedullin (ADM), atrial natriuretic peptide (ANP) and arginine vasopressin (AVP), and the fibrin degradation products D-dimer are associated with CAP and are under investigation in large cohorts for their role in diagnosis[1-6]. Among the biomarkers published, HLA-DR expression on circulating monocytes should be detected by flow cytometry, while the others are all serological markers. For urine analysis, the current reports mainly focus on urinary antigen tests or urinary metabolites as indicators of pneumonia.

We extracted the quantitative information about the above proteins from our quantitative proteomic studies (Fig. R1). In urine samples, we found that CRP (P02741) was significantly increased in COVID-19 patients and non-COVID-19 pneumonia patients compared to healthy controls (Fig. R1A).

HLA-DR showed a different profile. We detected both HLA-DRA and HLA-DRB1 in urine sample. We detected a significant increase of HLA-DRA (P01903) in non-COVID-19 patients compared to COVID-19 patients and healthy controls (Fig. R1B). The level of HLA-DRB1 (P01911) also increased significantly in non-COVID-19 patients compared to healthy controls and COVID-19 patients (Fig. R1C).

We did not detect PCT in urine samples, but we detected calcitonin. PCT is a prohormone of calcitonin. In physiological conditions, calcitonin is secreted by C cells of the thyroid gland, where it is generated from PCT. In the presence of infection, PCT is produced by macrophages and monocytic cells and released in the blood. PCT acts as a chemokine and modulates induction of anti-inflammatory cytokines [7]. We detected significantly decreased level of calcitonin in urine samples of non-COVID-19 patients compared to healthy controls, probably due to the dysregulation of PCT and the calcitonin secretion pathway. There was no significant difference between COVID-19 patients and healthy controls (Fig. R1D).

The blood level of D-dimer reflects the pathological role of coagulation and fibrinolysis in the development of acute lung injury, and thus it has been used as a biomarker of pneumonia[8]. Fibrinogen is converted into fibrin during coagulation. D-dimer is a degradation product which is generated when fibrin is digested by proteases. In our dataset, we detected fibrinogen alpha, beta and gamma chain but there was no significant difference between the levels of these three proteins among the three cohorts.

We also detected CD14 (P08571) and ADM (P35318) but with no significant changes. We did not detect IL-6, ANP and AVP in the urine samples.

In summary, we indeed detected some published biomarkers of pneumonia in our quantitative proteomic studies of urine samples from the different cohorts. CRP increased significantly in both COVID-19 and non-COVID-19 pneumonia, which is expected because CRP is an indicator of infection. However, CRP could not distinguish the COVID-19 and non-COVID-19 groups. HLA-DRA and HLA-DRB1 both increased significantly in non-COVID-19 pneumonia but to different levels. However, these proteins did not change in the urine samples of COVID-19 patients. Similarly, calcitonin decreased significantly in non-COVID-19 patients but showed no changes in COVID-19 patients compared to healthy controls. Thus, the level of HLA-DRA, HLA-DRB1 and calcitonin in urine samples could assist in distinguishing non-COVID-19 and COVID-19 pneumonia with other diagnostic tools.

Fig. R1. Quantitation of published biomarkers of CAP in the three different cohorts.

The scatter plot graphs showed four published potential diagnosis markers of CAP in the three different cohorts. Relative intensity is calculated by log (level of a certain protein/total protein level of the sample). One-way ANOVA was used to analyze the data. *p*-values are indicated by the asterisks as follows: ****, <0.0001; **, <0.01; *, <0.05, ns, no significant difference.

(3) Urine represents a highly selective filtrate of blood as a consequence of the activity of nephron activity. What markers were included to ensure that an altered proteomic profile in covid19 patients is or is not a consequence of an altered functioning of the nephrons? If available, the authors should report and correlate the proteomic data obtained to data that are a result of a standard clinical analysis of the urine. Specifically, are marker proteins that indicate normal functioning of nephrons affected during covid19 infection?

We thank the reviewer for making this important point which we agree with. The clinical characteristics of COVID-19 patients have been listed in the updated Table S1. We also revised the text in paragraph two, line 6-72, to *“We subdivided the 14 COVID-19 patients into 9 with moderate disease type and 5 with severe disease type according to the information provided by the attending physicians at the hospitals where the samples were obtained (Table S1).”*

Urine test results showed no proteinuria in six patients and mild (+) proteinuria in two patients. The clinical results showed no indication of renal impairment. No urine test results were available for the other six patients. These six samples came from a hospital in Wuhan. Under the emergency prevention and control of the pandemic, COVID-19 patients were admitted to the Department of Respiratory and Infectious Diseases and only viral nucleic acid, blood test and CT were required. Though we lack of some clinical data, we examined the expression level of albumin, hemoglobin and myoglobin in our proteome results. The presence of these proteins in the urine often indicates proteinuria of nephrotic patients. We found that none of these three proteins were significantly changed in urine samples.

Large amount of urinary proteins is one of the clinical symptoms of kidney function damage. Therefore, when proteomics is used to study kidney disease, a depletion of high abundant proteins is necessary during urine sample preparation process in order to avoid ion suppression effect. Even so, a large variation often appears in the results of protein identification. In our study, all proteomics experiments were done without depletion process and we identified about 2200 protein ID with a constant loading amounts of 300 ng for each sample. These results indicated that there is no ion suppression effect thus depletion for high abundant proteins are not necessary for our samples.

Taking these findings together, we conclude that there is no significant renal function loss in COVID-19 patients.

(4) The authors are carefully proposing a 2-stage model for covid19 progression based on the results obtained. The model lacks consistency in the sense that the first stage reflects molecular alterations (immune suppression and impaired tight junction) whereas the second stage reflects organ and organism wide changes. Thus, it is unclear whether it is really two stages of disease progression or merely a linear amplification of the molecular alterations that finally influence the functioning of whole organs. Indeed, specific alterations between early and late stage covid19 urine samples would be needed to support any two-stage model of disease.

We sincerely thank the reviewer for these very significant comments.

We further stratified the COVID-19 samples into subgroups in order to more explicitly elaborate the two-stage observation. All 14 COVID-19 patient samples in this study were collected in Wuhan, China, in February 2020 and they were firstly classified as mild to moderate according to the limited information back then. In view of the problem of mixed samples raised by the reviewer, we cross-checked with the attending physicians of the hospitals where the samples were obtained. According to more comprehensive clinical characteristics of the patients, based on the information provided by the doctors, the patients should be subdivided into 9 with moderate type and 5 with severe type (Updated Table S1). We also revised the text in paragraph two, line 6-72, to *“We subdivided the 14 COVID-19 patients into 9 with moderate disease type and 5 with severe disease type according to the information provided by the attending physicians at the hospitals where the samples were obtained (Table S1).”*

We performed the PCA for the two subgroups of COVID samples together with the healthy control group and the non-COVID group. The result showed a clear separation between these groups, which implies that the stratification may not affect our conclusions in this study (Fig. R2).

Fig. R2. Quality control of proteomic data.

(A) Cluster and (B) Principal Components Analysis (PCA) as evaluations for the data quality after standardization. Red color with “health” represents healthy control group; orange color with “moderate-nCoV” represents moderate stage of COVID-19 patient; blue color with “severe-nCoV” represents severe stage of COVID-19 patient; green color with “non-nCoV” represents non-COVID-19 lung infection patient group. A clear stratification shows a high quality of the data set.

We further analyzed the data for the moderate and severe subgroups. Gene Ontology enrichment analysis showed that the most of the upregulated proteins are involved in the complement and coagulation cascades, natural killer cell mediated cytotoxicity and platelet activation (Fig. R3A). Consistent with the previous result from the combined COVID-19 group, the moderate type group showed immune suppression (Fig. R3B), while the severe type group apparently showed immune activation compared with the moderate type group (Fig. R3C). This is consistent with literature reports of an excessive immune response and cytokine storm in patients in severe and critical stages of COVID-19. We also identified two characteristic proteins, Immunoglobulin lambda variable 3-25 (IGLV3-25) and Elongation factor 1-alpha 1 (EEF1A1), that may indicate the progression of the two stages of the COVID-19 infection (Fig. R3D).

Fig. R3. Immune system response in moderate and severe COVID-19 patients.
 A. Heatmap depicting the levels of differentially identified proteins in moderate and severe COVID-19 patients. The graphs show the relative intensity of differentially expressed proteins. Proteins included in the heatmap meet the requirement that fold change > 2 or < 0.5 and p value of less than 0.05 comparing severe to moderate patient samples.

- B. The interaction diagram of proteins involved in the innate immune response, response to virus, antigen processing and presentation and T cell activation. Network nodes and edges represent proteins and protein-protein associations. Green solid line represents inhibition, red solid line represents activation, gray dotted line represents GO pathway. Color bar from red to green represents the fold change of protein level from increasing to decreasing.
- C. The interaction diagram of proteins involving in antigen processing and presentation, complement activation, cellular response to chemokine, regulation of immune response, T cell activation, T cell receptor signaling pathway. Green solid line represents inhibition, red solid line represents activation, gray dotted line represents GO pathway. Color bar from red to green represents the fold change of protein level from increasing to decreasing.
- D. The scatter plot graph showed two indicate proteins which are potential diagnosis markers for severe COVID-19 patients. One-way ANOVA was used to analyze the data. *p*-values are indicated by the asterisks as follows: ****, <0.0001, **, <0.01.

Our new data provide stronger evidence that two distinct immune responses appear in the two disease stages of COVID-19. The immune response was suppressed in the early stage of infection (moderate type), and immune activation and excessive immune response emerged in the severe stage of infection. This result further verified the two-stage model we proposed in the manuscript. We added Fig. R4 to the supplementary figure of as New Fig. S3D in revised manuscript. We have added Fig. R3 to the main figure as New Fig. 3 and the previous Fig. 3 was changed to New Fig. 4 in revised manuscript.

We rewrote the text in paragraph seven, line 192-208, to “*It is reported that a cytokine storm happens at the late stage of COVID-19 patients¹⁴⁻¹⁶. To understand the pathogenesis of COVID-19, it will be essential to find out how the transitions take place during the progression of the disease. To further investigate this, we subdivided the COVID-19 patients into 9 moderate cases and 5 severe cases. PCA revealed a good separation of the moderate and severe COVID-19 patient samples (Fig.S3 C, D). Gene Ontology enrichment analysis showed that the most of the upregulated proteins are involved in the complement and coagulation cascades, natural killer cell mediated cytotoxicity and platelet activation (Fig 3A). Looking in detail an the moderate and severe subgroups, we found, interestingly, that an activated immune response emerged to a certain extent in the late stage of the disease while the immunosuppression effect remained in the early stage. (Fig3 B-C, Fig. S11-13). This results indicate that in the late stage of the disease the immune response was activated which is in consistent with an excessive immune response and cytokine storm in patients in severe and critical stages of COVID-19 patients¹⁴⁻¹⁶. Our study also identified two characteristic proteins, Immunoglobulin lambda variable 3-25 (IGLV3-25) and Elongation factor 1-alpha 1 (EEF1A1), that may indicate the progression of the two stages of the COVID-19 disease. (Fig 3D).*”

We revised the text in paragraph eight, line 229-234, to “*In conclusions, we applied the most advanced mass spectrometry technology to perform quantitative proteomics analysis of urine samples from COVID-19 patients and healthy controls and non-COVID-19 pneumonia patients. We uncovered changes in the protein landscape that revealed immunosuppression and impaired tight junctions in COVID-19 patients in the early stage of infection. Intriguingly, we also detected immune activation to a certain extent in the late stage of infection (Fig. 4).*”

Reference:

1. Leoni, D. and J. Rello, *Severe community-acquired pneumonia: optimal management*. Curr Opin Infect Dis, 2017. **30**(2): p. 240-247.
2. Karakioulaki, M. and D. Stolz, *Biomarkers in Pneumonia-Beyond Procalcitonin*. Int J Mol Sci, 2019. **20**(8).
3. Menendez, R., et al., *Stability in community-acquired pneumonia: one step forward with markers?* Thorax, 2009. **64**(11): p. 987-92.
4. Andrijevic, I., et al., *Interleukin-6 and procalcitonin as biomarkers in mortality prediction of hospitalized patients with community acquired pneumonia*. Ann Thorac Med, 2014. **9**(3): p. 162-7.
5. Burgos, J., et al., *Determination of neutrophil CD64 expression as a prognostic biomarker in patients with community-acquired pneumonia*. Eur J Clin Microbiol Infect Dis, 2016. **35**(9): p. 1411-6.
6. Zhuang, Y., et al., *Predicting the Outcomes of Subjects With Severe Community-Acquired Pneumonia Using Monocyte Human Leukocyte Antigen-DR*. Respir Care, 2015. **60**(11): p. 1635-42.
7. Prucha, M., G. Bellingan, and R. Zazula, *Sepsis biomarkers*. Clin Chim Acta, 2015. **440**: p. 97-103.
8. Olson, J.D., *D-dimer: An Overview of Hemostasis and Fibrinolysis, Assays, and Clinical Applications*. Adv Clin Chem, 2015. **69**: p. 1-46.

Reviewer #3

This was a focused proteomics analysis of urine from three classes of subjects: healthy subjects, COVID-19 patients and patients with non-COVID pneumonia. Based on proteomics combined with KEGG analysis, they found that in COVID patients there was protein content signature that suggested early suppression of the immune response (Figure 2A) and that there was an enhancement of tight junction proteins present in urine (Figure 2B). Based on their analysis, they determined that COVID-19 patients showed hallmark changes of a set of 8 proteins associated with a COVID-specific response (e.g. increased β 2M, TRIM4; decreased PRKACA, LPL), which may have diagnostic value.

We thank the reviewer for recognizing our work.

The authors proposed that the protein expression profile suggests that “the COVID-19 virus may alter intercellular TJ formation and epithelial morphogenesis during viral invasion through the human airway.” (Lines 149-151). However, with the exception of claudin-3, the claudin profile they identified from the urine samples (claudin-2, claudin-11, claudin-19) does not match the expected claudin expression profile if these proteins were to have originated in the lung epithelium (e.g. high claudin-4, claudin-7, claudin-18). Based on this it is not clear whether the proteins represent damaged lung epithelial cells or whether they are due to damage from other cells.

We sincerely thank the reviewer for the comments and suggestions. We agree with the reviewer that it is not possible to state that the tight junction proteins were specifically all from lung epithelium tissues without further validation. Indeed, the urine proteome represents the overall result of the body infected by the virus. The identified proteins can be from other organ/tissue cells as well. We further used electron microscopy to observe the tight junctions of the small intestinal tissue of ACE2 transgenic mice infected by SARS-Cov2 virus. We found that, compared to the control group, the length of the tight junction in small intestinal mucosal epithelium was significantly shortened in the viral infection group (Fig. R4). In addition, claudin-3 and CGN, which were identified to be decreased in urine samples, were highly expressed in small intestine. The small intestine was reported that it expresses ACE2 at high levels and can be infected by the SARS-Cov2 virus [1, 2]. This result also provides evidence that the tight junction proteins are likely other organs/tissues in addition to the lungs.

Fig. R4. Tight junction length changes of small intestinal epithelium of ACE2 transgenic mice (A) without infection as a control and (B) infected by SARS-COV-2 virus. Red double arrows indicate the position of the tight junction. Scale bar is 500 nm.

(C) Scatter plot graph to quantify the tight length changes of small intestinal epithelium of ACE2 transgenic mice without (as a control) and with SARS-COV-2 infection. T test was used to analyze the data. *p*-value is indicated by the asterisks as follows: ****, <0.0001.

We revised the text in paragraph five, line 154-155, to “*Tight junctions also exist between epithelial cells in other organs, such as intestine, kidney and brain*16,17”.

Another concern is that the samples were combined COVID-19 patients with mild-to-moderate stage disease; it would strengthen the study to stratify these samples into two different groups and to also analyze severe stage COVID-19 patients. Also, the type of pneumonia of the non-COVID patients was not clear (e.g. bacterial vs viral) and was also combined. Although the PCA in Figure S3 analysis suggests that the three different groups of patients do not overlap, there may still be subgroups within each set of subjects.

Although the proteomics analysis was interesting, the study as it currently stands is limited and does not provide experimental validation of the model of the progression of COVID-19 proposed in Figure 3.

We sincerely thank the reviewer for the very constructive suggestion of further stratifying the samples into subgroups.

We further stratified the COVID-19 samples into subgroups in order to more explicitly elaborate the two-stage observation. All 14 COVID-19 patient samples in this study were collected in Wuhan, China, in February 2020 and they were firstly classified as mild to moderate according to the limited information back then. In view of the problem of mixed samples raised by the reviewer, we cross-checked with the attending physicians of the hospitals where the samples were obtained. According to more comprehensive clinical characteristics of the patients, based on the information provided by the doctors, the patients should be subdivided into 9 with moderate type and 5 with severe type (Updated Table S1). Unfortunately that we could not obtain more information of the patients in the pneumonia non-COVID group, therefore, we could not further stratify these samples.

We revised the text in paragraph two, line 60-72, to “*We subdivided the 14 COVID-19 patients into 9 with moderate disease type and 5 with severe disease type according to the information provided by the attending physicians at the hospitals where the samples were obtained (Table S1).*” line 78-77, to “*For the data analysis, we firstly compared the three groups of COVID-19 infection cases, healthy donors and non-COVID-19 pneumonia cases.*”

We performed the PCA for the two subgroups of COVID samples together with the healthy control group and the non-COVID group. The result showed a clear separation between these groups, which implies that the stratification may not affect our conclusions in this study (Fig. R2).

Fig. R2. Quality control of proteomic data.

(A) Cluster and (B) Principal Components Analysis (PCA) as evaluations for the data quality after standardization. Red color with “health” represents healthy control group; orange color with “moderate-nCoV” represents moderate stage of COVID-19 patient; blue color with “severe-nCoV” represents severe stage of COVID-19 patient; green color with “non-nCoV” represents non-COVID-19 lung infection patient group. A clear stratification shows a high quality of the data set.

We further analyzed the data from the moderate and severe subgroups. Gene Ontology enrichment analysis showed that the most of the upregulated proteins are involved in the complement and coagulation cascades, natural killer cell mediated cytotoxicity and platelet activation (Fig. R3A). Consistent with the previous results from the combined COVID-19 group, the moderate type group also showed immune suppression (Fig. R3B), while the severe type group apparently showed immune activation compared with the moderate type group (Fig. R3C). This is consistent with literature reports of an excessive immune response and cytokine storm in patients in severe and critical stages of COVID-19. We also identified two characteristic proteins, Immunoglobulin lambda variable 3-25 (IGLV3-25) and Elongation factor 1-alpha 1 (EEF1A1), that may indicate the progression of the two stages of the COVID-19 infection (Fig. R3D).

Fig. R3. Immune system response in moderate and severe COVID-19 patients.
 A. Heat map depicting the levels of differentially identified proteins in moderate and severe COVID-19 patients. The graphs show the relative intensity of differentially expressed proteins. Proteins included in the heatmap meet the requirement that fold change > 2 or < 0.5 and p value of less than 0.05 comparing severe to moderate patient samples.

- B. The interaction diagram of proteins involved in the innate immune response, response to virus, antigen processing and presentation and T cell activation. Network nodes and edges represent proteins and protein-protein associations. Green solid lines represent inhibition; gray dotted lines represent GO pathways. Color bar from red to green represents the fold change of protein level from increasing to decreasing.
- C. The interaction diagram of proteins involving in antigen processing and presentation, complement activation, cellular response to chemokine, regulation of immune response, T cell activation, T cell receptor signaling pathway. Green solid lines represent inhibition; red solid lines represent activation; gray dotted lines represent GO pathways. Color bar from red to green represents the fold change of protein level from increasing to decreasing.
- D. The scatter plot graphs showing two proteins which are potential diagnostic markers for severe COVID-19. One-way ANOVA was used to analyze the data. P-values are indicated by the asterisks as follows: ****, <0.0001; **, <0.01.

We sincerely thank the reviewer for pointing out this significant point of stratification. Our new data provide stronger evidence that two distinct immune responses appear in the two disease stages of COVID-19 patients. The immune response was suppressed in the early stage of infection (moderate type), and the immune activation and excessive immune response emerged in the severe stage of infection. This result further verified the two-stage model we proposed in the manuscript. We added Fig. R2 to the supplementary figure of as New Fig. S3D in revised manuscript. We added Fig. R3 to the main figure as New Fig. 3 and the previous Fig. 3 was changed to New Fig. 4 in revised manuscript.

In order to state our conclusion more accurately, we changed the title of the manuscript to “*Immune suppression in the early stage of COVID-19 disease*” and also rewrote the text in paragraph seven, line 192-208, to “*It is reported that a cytokine storm happens at the late stage of COVID-19 patients14-16. To understand the pathogenesis of COVID-19, it will be essential to find out how the transitions take place during the progression of the disease. To further investigate this, we subdivided the COVID-19 patients into 9 moderate cases and 5 severe cases. PCA revealed a good separation of the moderate and severe COVID-19 patient samples (Fig.S3 C, D). Gene Ontology enrichment analysis showed that the most of the upregulated proteins are involved in the complement and coagulation cascades, natural killer cell mediated cytotoxicity and platelet activation (Fig 3A). Looking in detail an the moderate and severe subgroups, we found, interestingly, that an activated immune response emerged to a certain extent in the late stage of the disease while the immunosuppression effect remained in the early stage. (Fig3 B-C, Fig. S11-13). This results indicate that in the late stage of the disease the immune response was activated which is in consistent with an excessive immune response and cytokine storm in patients in severe and critical stages of COVID-19 patients14-16. Our study also identified two characteristic proteins, Immunoglobulin lambda variable 3-25 (IGLV3-25) and*

Elongation factor 1-alpha 1 (EEF1A1), that may indicate the progression of the two stages of the COVID-19 disease. (Fig 3D).”

References:

1. Ziegler, C.G.K., et al., *SARS-CoV-2 Receptor ACE2 Is an Interferon-Stimulated Gene in Human Airway Epithelial Cells and Is Detected in Specific Cell Subsets across Tissues*. *Cell*, 2020. **181**(5): p. 1016-1035 e19.
2. Li, M.Y., et al., *Expression of the SARS-CoV-2 cell receptor gene ACE2 in a wide variety of human tissues*. *Infect Dis Poverty*, 2020. **9**(1): p. 45.

REVIEWERS' COMMENTS

Reviewer #2 (Remarks to the Author):

Thank you for the detailed answers in the response to the reviewers' comments.

Major comment: Please deposit the data in a common data repository like ProteomeXchange. Proteomic data repositories allow for uploading the data prior to publication and the data can be made available for the reviewing process.

Major comment: The response to the reviewers reads very well and detailed. Please include the paragraphs from the response in the main body of the text because sometimes individual statements in the main body of the manuscript remain unexplained but are well explained in the detailed response to the reviewers.

Thank you for including the clinical characteristics in table S1b.

Further comments on the main body of the text:

l35ff: Please carefully check spelling and grammar in the abstract. First sentence might better be placed in present tense.

l40f: "unusual covert virus infections" needs more explanation or an extension to make clear what exactly "covert" means in this context.

l75ff: citation: "Our data were acquired with high quality according to the Principal Components Analysis (PCA) (Fig. S3)." Comment: PCA is not a measure of data quality but a statistical tool to assess whether systematic differences between datasets exist. Please choose an alternative method to assess data quality. Options are to report reproducibility across ms runs, the consistence of data obtained between runs or equivalent.

l82ff: the use of the term "down-regulation" is confusing: the authors show that there is less protein in urine which is a body's filtrate, not that it is down regulated in expression level in cells for example. The point is that it remains unclear where and when "regulation" comes from: Is it the altered activity of nephrons, a passive failure of filtering the blood, or is it an active shut down of the filtering process which would then be regulated? Or is it gene expression changes in a completely different cell type or tissue? - The authors can sort this out in part based on the abundance levels of known urine protein markers.

l128f: "...suggests the significant impairment of the complement system (Fig. S7)." Might be better placed into discussion because the question that arises is whether this observation is supported by reduced Complement protein levels in the blood of COVID19 patients?

l1301ff: this interpretation might be better placed into discussion: "...indicating that the phagocytosis of microphages, neutrophils, natural killer cells and monocytes was suppressed...". Is it a suppression of phagocytosis or a reduced number of macrophages (pls. correct typo in the manuscript), neutrophils, nkc's and monocytes in blood and, consequently, proteins of these cells in urine? In other words, are other proteins that are markers for these cell types unaltered or also altered in urine? - Fig.2A indicates that additional, nkc associated proteins are also less abundant suggesting absence of these cells rather than a reduced activity of the cells.

l150f: reference is missing.

l187ff: It is interesting that new biomarkers might emerge from this study. Can the authors describe their findings in more detail? The data suggests that some of the proteins proposed as

biomarkers are also significantly different in abundance in urine between healthy and non-nCOV. Thus to which degree allow these proteins for a differential diagnosis in the context presented here. Are these proteins identified as urine biomarkers for other diseases as well?

Reviewer #3 (Remarks to the Author):

This is a revised version of a focused proteomics analysis of urine from three classes of subjects: healthy subjects, COVID-19 patients and patients with non-COVID pneumonia. In response to the previous critiques the authors further stratified the data from COVID-19 based on severity. They also made a more general statement about the origin of tight junction proteins being due to general tissue damage as opposed to specific to the lung, which is a more accurate statement of the conclusions of the study. The authors also ruled out overt kidney failure as a potential source of differential proteins present in urine.

Based on proteomics combined with KEGG analysis, they found that in COVID patients there was protein content signature that suggested early suppression of the immune response (Figure 2A) and that there was an enhancement of tight junction proteins present in urine (Figure 2B). Based on their analysis, they determined that COVID-19 patients showed hallmark changes of a set of 8 proteins associated with a COVID-specific response (e.g. increased β 2M, TRIM4; decreased PRKACA, LPL), which may have diagnostic value.

Of note, the new data in Figure 3 showing that T cell responses predominate over innate immunity in severely ill COVID-19 patients supports a two stage model for pathogenesis. Still the dataset is correlative as opposed to longitudinal which limits the ability to fully link their findings to the progression and severity of the disease.

REVIEWERS' COMMENTS

We thank all reviewers for their approving our revision.

Reviewer #2 (Remarks to the Author):

Thank you for the detailed answers in the response to the reviewers' comments.

We thank the reviewer for approving our revision.

Major comment: Please deposit the data in a common data repository like ProteomeXchange. Proteomic data repositories allow for uploading the data prior to publication and the data can be made available for the reviewing process.

We added the section of Data availability as below:

Reference FASTA files contain human UNIPROT database (only reviewed entries) (human 20,421 entries, downloaded July 2019) and SARS-CoV-2 Uniprot database which combined with SARS protein database (38 viral entries, April 2020). Latest GO database (<https://www.ebi.ac.uk/QuickGO/>) and KEGG pathway database (<https://www.kegg.jp/kegg/pathway.html>) were used for gene ontology and pathway enrichment analysis. The KEGG ligand database (<https://www.kegg.jp/kegg/ligand.html>) was used to obtain the compound and enzyme relationship. The experimental data that support the findings of this study have been deposited in iProX (integrated proteome resources) of ProteomeXchange with the accession code PXD020522. The data could be accessed from <https://www.iprox.org/page/SSV024.html?url=1601217952947riX4> with Extraction password of 3Efy. The source data underlying Fig. 1c, Fig. 2ab, Fig. 2c, Fig. 3ac, Fig. 3b, Fig. 3d are provided as a Source Data file. Source data are provided with this paper.

Major comment: The response to the reviewers reads very well and detailed. Please include the paragraphs from the response in the main body of the text because sometimes individual statements in the main body of the manuscript remain unexplained but are well explained in the detailed response to the reviewers.

We thank the reviewer for approving our response. We have included some of the response in the main body of the text.

Thank you for including the clinical characteristics in table S1b.

We thank the reviewer for approving our revision of the clinical characteristics.

Further comments on the main body of the text:

l35ff: Please carefully check spelling and grammar in the abstract. First sentence might better be placed in present tense.

We revised the text to “We report a quantitative proteomic analysis of urine samples from COVID-19 infection cases, healthy donors and non-COVID-19 pneumonia cases.”

l40f: "unusual covert virus infections" needs more explanation or an extension to make clear what exactly "covert" means in this context.

We revise the text to “unusual virus infections”.

l75ff: citation: "Our data were acquired with high quality according to the Principal Components Analysis (PCA) (Fig. S3)." Comment: PCA is not a measure of data quality but a statistical tool to assess whether systematic differences between datasets exist. Please choose an alternative method to assess data quality. Options are to report reproducibility across ms runs, the consistence of data obtained between runs or equivalent.

We thank the reviewer for pointing out this. We have revised the main text “Our data were acquired with high quality according to the Principal Components Analysis (PCA)” to “Our data showed clear stratification among different cohorts according to the Principal Components Analysis (PCA).”

We have revised the figure caption of Fig.S3 as below:

Fig. S3. Inter-group difference analysis.

(a. b) Cluster (a) and Principal Components Analysis (PCA) (b) showed the inter-group difference before standardization. Blue color with “health” represents healthy control group; orange color with “nCoV” represents COVID-19 patient group; green color with “non-nCoV” represents non-COVID-19 lung infection patient group. A clear stratification shows that the three groups can be well distinguished.

(c. d) Cluster (c) and Principal Components Analysis (PCA) (d) showed the inter-group difference after standardization. Red color with “health” represents healthy control group; orange color with “moderate-nCoV” represents moderate stage of COVID-19 patient; blue color with “severe-nCoV” represents severe stage of COVID-19 patient; green color with “non-nCoV” represents non-COVID-19 lung infection patient group. A clear stratification shows that the four groups can be well distinguished. nCOV group can be further divided into Moderate-nCoV and Severe-nCoV group, respectively.

Each patient sample was a biological duplicate. A mixed aliquot from each sample served as quality control (QC) sample was applied after every four runs. The median coefficient of variation (CV) for qualification was 18.6% on the protein level after median normalization. It presents a good quality of the dataset.

l82ff: the use of the term "down-regulation" is confusing: the authors show that there is less protein in urine which is a body's filtrate, not that it is down regulated in expression level in cells for example. The point is that it remains unclear where and when "regulation" comes from: Is it the altered activity of nephrons, a passive failure of filtering the blood, or is it an active shut down of the filtering process which would then be regulated? Or is it gene expression changes in a completely different cell type or tissue? - The authors can sort this out in part based on the abundance levels of known urine protein markers.

l128f: "...suggests the significant impairment of the complement system (Fig. S7)." Might be better placed into discussion because the question that arises is whether this observation is supported by reduced Complement protein levels in the blood of COVID19 patients?

l1301ff: this interpretation might be better placed into discussion: "...indicating that the phagocytosis of microphages, neutrophils, natural killer cells and monocytes was suppressed...". Is it a suppression of phagocytosis or a reduced number of macrophages (pls. correct typo in the manuscript), neutrophils, nkc's and monocytes in blood and, consequently, proteins of these cells in urine? In other words, are other proteins that are markers for these cell types unaltered or also altered in urine? - Fig.2A indicates that additional, nkc associated proteins are also less abundant suggesting absence of these cells rather than a reduced activity of the cells.

We thank the reviewer for these three questions. These questions are correlated and could be answered together.

Whether urine proteome can reflect changes in blood or intracellular protein levels has been a common concern in the field of urinary proteomics. Other research groups have shown that the urinary proteome is, to some extent, a reflection of blood and intracellular protein levels [1-2]. Most urine proteins are derived from plasma components that cross the glomerular filtration barrier and those released from the kidneys and the urinary tract. Thus, in the absence of primary urinary disease, the protein composition of urine is a good indicator of the health status of the body. In recent years, urine has been widely used in disease proteomic analysis and as an important source of clinical samples for early warning and detection of diseases due to its non-invasive and easy access. In this research, there is none of our enrolled patients had been diagnosed with clinical renal injury, and yet we identified large-scale protein differences in urine. Thus, we reasoned that the “decreased level of protein” in urine mostly reflects the “down-regulation” of the protein in other cell type or tissue.

The paragraph below has been added to the main body of the text:

“Several studies have shown that the compositions of proteins detected in urine samples can genuinely reflect the changes of the body health condition. The majority of urinary proteins originate from plasma components that pass through the glomerular filtration barrier, as well as liberated proteins from the kidney and urinary tract. Thus, in the absence of primary urological disease, the protein composition of urine is an appropriate mirror of general health status.”

1. Zhao, M., et al., *A comprehensive analysis and annotation of human normal urinary proteome*. Sci Rep, 2017. **7**(1): p. 3024.
2. Wu, J. and Y. Gao, *Physiological conditions can be reflected in human urine proteome and metabolome*. Expert Rev Proteomics, 2015. **12**(6): p. 623-36.

l150f: reference is missing.

Thank you for this good point. We have added the reference in the main text as 19-21: “It is reported that a cytokine storm happens at the late stage of COVID-19 patients¹⁹⁻²¹”

l187ff: It is interesting that new biomarkers might emerge from this study. Can the authors describe their findings in more detail? The data suggests that some of the proteins proposed as biomarkers are also significantly different in abundance in urine between healthy and non-nCOV. Thus to which degree allow these proteins for a differential diagnosis in the context presented here. Are these proteins identified as urine biomarkers for other diseases as well?

We thank the reviewer for this question.

In our study in Figure 2C, we identified 6 proteins which are differentially expressed when comparing COVID-19 with non-nCov patients. These proteins also distinguish COVID-19 with health control. Thus, these 6 proteins may be eligible to differentially diagnose COVID-19 and non-nCov patients. However, for differentiation between healthy control and non-nCov patients, only 3 out of these 6 identified proteins showed significance, other 3 showed no changes. We apologize for this confusion.

The identified proteins all have very important functions in a variety of disease conditions and some have potential in diagnosis in other diseases. ApoA4 is potential early diagnostic biomarkers for prediabetes, liver fibrosis, and ovarian cancer(1); Abnormalities in LPL function associated with a number of pathophysiological conditions, including atherosclerosis, chylomicronaemia, obesity, Alzheimer's disease, and dyslipidaemia associated with diabetes, insulin resistance, and infection(2). The upregulation of MT1G together with other MTs is related to poor prognosis of large cell lung cancer, drug resistance in Gastric carcinoma(3). FOLR2 are highly expressed on cancer cells and activated macrophages, indicating that FOLR2 is a potential marker for diagnosis of cancers and activated macrophage-mediated inflammatory diseases(4); In all tested samples in this study, patients did not show renal dysfunction. Therefore, raised beta 2-microglobulin level could reflect serum levels of beta 2-microglobulin, which is related to malignancy, neoplastic proliferation of lymphoid B-cells or in inflammatory disorders connected with an activation of the lymphopoietic system(5).

Thus, based on our current data, we would reason that the proteins identified in this study could be used as differential diagnosis marker to make it more precise. We will expand the sample size and further validate the potential of these markers identified in current study.

We change the main text “These proteins are potential biomarkers for differential diagnosis of COVID-19” to “These proteins are potential biomarkers for differential diagnosis of COVID-19 to make the it more precise.”

1. M. Si, J. Lang, The roles of metallothioneins in carcinogenesis. *J Hematol Oncol* **11**, 107 (2018).
2. J. Qu, C. W. Ko, P. Tso, A. Bhargava, Apolipoprotein A-IV: A Multifunctional Protein Involved in Protection against Atherosclerosis and Diabetes. *Cells* **8**, (2019).
3. J. R. Mead, S. A. Irvine, D. P. Ramji, Lipoprotein lipase: structure, function, regulation, and role in disease. *J Mol Med (Berl)* **80**, 753-769 (2002).
4. Y. S. Yi, Folate Receptor-Targeted Diagnostics and Therapeutics for Inflammatory Diseases. *Immune Netw* **16**, 337-343 (2016).
5. R. E. Turnham, J. D. Scott, Protein kinase A catalytic subunit isoform PRKACA; History, function and physiology. *Gene* **577**, 101-108 (2016).

Reviewer #3 (Remarks to the Author):

This is a revised version of a focused proteomics analysis of urine from three classes of subjects: healthy subjects, COVID-19 patients and patients with non-COVID pneumonia. In response to the previous critiques the authors further stratified the data from COVID-19 based on severity. They also made a more general statement about the origin of tight junction proteins being due to general tissue damage as opposed to specific to the lung, which is a more accurate statement of the conclusions of the study. The authors also ruled out overt kidney failure as a potential source of differential proteins present in urine.

We thank the reviewer for approving our revision.

Based on proteomics combined with KEGG analysis, they found that in COVID patients there was protein content signature that suggested early suppression of the immune response (Figure 2A) and that there was an enhancement of tight junction proteins present in urine (Figure 2B). Based on their analysis, they determined that COVID-19 patients showed hallmark changes of a set of 8 proteins associated with a COVID-specific response (e.g. increased β 2M, TRIM4; decreased PRKACA, LPL), which may have diagnostic value.

We thank the reviewer for approving our revision.

Of note, the new data in Figure 3 showing that T cell responses predominate over innate immunity in severely ill COVID-19 patients supports a two stage model for pathogenesis. Still the dataset is correlative as opposed to longitudinal which limits the ability to fully link their findings to the progression and severity of the disease.

We thank the reviewer for pointing out this point. We incorporated this comments into section of Discussion: “The limitation of this study is that the dataset is correlative but not longitudinal. Nevertheless, these unusual features of COVID-19 during the course of human infections will guide us to further understanding of COVID-19 pathogenesis, mechanistic study and clinical treatments.”